# Restoration of Neurological Function Following Peripheral Nerve Trauma

**DOI:** 10.3390/ijms21051808

**Published:** 2020-03-06

**Authors:** Damien P. Kuffler, Christian Foy

**Affiliations:** 1Institute of Neurobiology, Medical Sciences Campus, University of Puerto Rico, 201 Blvd. del Valle, San Juan, PR 00901, USA; 2Section of Orthopedic Surgery, Medical Sciences Campus, University of Puerto Rico, San Juan, PR 00901, USA; christian.foy@upr.edu

**Keywords:** nerve repair, nerve gaps, platelet-rich plasma (PRP), platelet-rich plasma, axon regeneration, allografts, autografts, nerve conduits

## Abstract

Following peripheral nerve trauma that damages a length of the nerve, recovery of function is generally limited. This is because no material tested for bridging nerve gaps promotes good axon regeneration across the gap under conditions associated with common nerve traumas. While many materials have been tested, sensory nerve grafts remain the clinical “gold standard” technique. This is despite the significant limitations in the conditions under which they restore function. Thus, they induce reliable and good recovery only for patients < 25 years old, when gaps are <2 cm in length, and when repairs are performed <2–3 months post trauma. Repairs performed when these values are larger result in a precipitous decrease in neurological recovery. Further, when patients have more than one parameter larger than these values, there is normally no functional recovery. Clinically, there has been little progress in developing new techniques that increase the level of functional recovery following peripheral nerve injury. This paper examines the efficacies and limitations of sensory nerve grafts and various other techniques used to induce functional neurological recovery, and how these might be improved to induce more extensive functional recovery. It also discusses preliminary data from the clinical application of a novel technique that restores neurological function across long nerve gaps, when repairs are performed at long times post-trauma, and in older patients, even under all three of these conditions. Thus, it appears that function can be restored under conditions where sensory nerve grafts are not effective.

## 1. Introduction

Clinically, traumatic peripheral nerve injuries are common, and are caused by violence, recreational activities, motor vehicle accidents, and iatrogenic injuries during surgery. The majority of nerve injuries occur in the upper extremity [1] with about 1–3% of all upper extremity trauma patients presenting with nerve injuries [2]. These injuries can be severely debilitating and have a significantly negative impact on the individual’s lifestyle, function, and work [3,4]. The majority of those who suffer traumatic nerve injuries are young, with an average age of 39 [5]. Less than 50% of such individuals undergo nerve repair surgery, and of those who do, only 40–50% recover good function [6]. Thus, the majority of individuals who suffer peripheral nerve traumas suffer permanent neurological deficits, and frequently also chronic neuropathic pain associated with the nerve injury.

Due to the generally limited extent of neurological recovery, it is essential to develop novel techniques that restore more extensive function to a larger number of patients. This review examines the relative efficacies of different techniques that have been tested for their ability to restore function and discusses a novel technique that shows great promise for inducing recovery under conditions where it is presently not possible.

## 2. Issues Influencing the Extent of Axon Regeneration and Neurological Recovery

### 2.1. Type of Nerve Trauma

The type of nerve injury has a major influence on the extent of neurological recovery. Following a traumatic nerve injury, sometimes referred to as an “untidy” wound (shrapnel, bullet, blunt object, open fracture, contaminated), there is significantly less recovery than following a “tidy,” or clean-cut, injury (glass, knife, scissors) [7]. This is because untidy injuries damage longer lengths of the nerve, which must be removed, resulting in longer nerve gaps, from which recovery is less than for short gaps.

Another type of nerve injury involves the loss of nerve vascularization, such as occurs with untidy injuries. When nerves lose their blood supply, a significantly lower percentage of those repaired nerves recover function than those following the repair of a tidy injury [7]. This is because of the time required for the nerve graft to be re-vascularized, during which axons do not, or are permanently prevented from, regenerating into the non-vascularized portion of the nerve.

### 2.2. Gender and Age

Although some behavioral differences are seen in neurons and Schwann cells associated with gender and age, their impact on clinical neurological recovery following nerve trauma is not clear. Thus, it has been observed that the density of epidermal nerve fibers decreases with age, and is lower in men compared with women [8]. This suggests age and gender influence innervation.

In hamsters, recovery of function following facial nerve injury is significantly faster in females than males [9]. While the administration of exogenous steroids accelerates axon regeneration in males, it has a lower impact on the rate of regeneration in females [9]. In rats, continuous exercise training increases the extent of axon regeneration in male, but not female, or castrated rats [10,11]. However, interval exercise training enhances axon regeneration in female, but not male rats [10,11].

In the male animal model, the influence on axon regeneration is associated with the expression of androgens [12]. They influence the extent of axon regeneration in peripheral nerves by regulating motor neurons’ expression of brain-derived neurotrophic factor (BDNF) and its receptor, trkB [11,13,14]. These molecules, in turn, influence the extent of axon regeneration [10,12]. The effect of training on axon regeneration in females works through a different mechanism than that of males [12]. Further studies are required to determine whether hormone treatment strategies may be effective in enhancing the extent of neurological recovery following nerve injury.

### 2.3. Promoting Axon Regeneration Through Crushed Nerves

Two to three days after a crush nerve injury, the severed axons begin to regenerate into the distal part of the nerve and continue to regenerate until they reach and reinnervate their original targets. Axon regeneration is promoted by the denervated Schwann cells in the distal portion of the nerve by their release of neurotrophic factors, and their extracellular matrix [15,16,17,18,19,20,21]. The greater the number of axons that regenerate through the distal nerve, the greater the extent of neurological recovery [19,22] Generally, the precision of target reinnervation is extremely high [19].

### 2.4. Restoration of Function without Surgical Intervention

Following a nerve transection, the nerve stumps normally retract, resulting in a gap of ≤3 mm. Despite this small gap, neurological recovery may develop without surgical intervention. This is due to a cascade of events involving the diffusion of fibrinogen from leaky blood vessels into the nerve gap where it combines with thrombin. This leads to fibrinogen polymerization and the formation of a three-dimensional fibrin matrix within the gap [23]. This matrix provides passive support to axons, which allows them to regenerate to the distal nerve stump.

However, fibrin clots are converted into a matrix that actively promotes axon regeneration by the migration of Schwann cells into the fibrin clot from the central and distal nerve stumps. These Schwann cells release a cocktail of neurotrophic and wound healing factors that bind to the pure fibrin converting it from a passive three-dimensional matrix into one that actively promotes axon regeneration [24]. This results in a significant increase in the number of axons that regenerates across the gap [25].

The efficacy of fibrin in promoting axon regeneration is increased by the platelets and mesenchymal stem cells that become entrapped in the fibrin clot in the nerve gap. They act by multiple mechanisms: (1) They release neurotrophic and other factors that act directly on the axons to promote regeneration [26]. (2) They release factors that promote Schwann cells of the distal nerve pathway to proliferate and release neurotrophic factors, which also enhance the extent of axon regeneration [27]. (3) The mesenchymal stem cells differentiate into Schwann cells, which release neurotrophic and other factors, thus enhancing the concentration of these factors and the extent of axon regeneration [28]. (4) Mesenchymal stem cells release factors that induce angiogenesis, which is essential for axon regeneration [29].

Although factors within the fibrin clot promote axon regeneration, factors are also required to direct axons across the nerve graft. Growth cones extend fine processes that sample the environment around them in search of factors to which they can adhere, and that both promote and direct their growth. When Schwann cell-released neurotrophic factors are distributed in a uniform concentration around neurons and their growth cones, neurite outgrowth is promoted, but the outgrowth is random. However, when neurons and their growth cones are exposed to a concentration gradient of those same factors in vitro and in vivo, the growth cones turn and increase the concentration gradient of the Schwann cell-released factors [19,30]. This is because, as the factors diffuse away from the distal nerve stump, they create a concentration gradient of the factor, which is the highest at the distal nerve stump [30,31,32,33]. This directs the axons up the gradient and to the distal stump [19,30,33,34,35] Once the axons reach the distal nerve stump, their regeneration continues to be promoted and directed through the distal nerve segment by the concentration gradient of Schwann cell-released factors ahead of them.

### 2.5. Restoration of Function with Surgical Intervention—Anastomosis

When a nerve has a clean transection, or when the nerve defect is small (only a few millimeters), the nerve stumps can be anastomosed, which leads to the restoration of function. However, to develop functional recovery, the repaired nerve must be tension-free [36]. When anastomosis is performed within 14 days of nerve trauma, functional recovery is good in about 80% of patients [37]. However, with increasing time between nerve trauma and anastomosis, the extent of recovery decreases [38,39,40]. The types of changes that occur over time that lead to this decrease in recovery are discussed below.

### 2.6. Promoting Axon Regeneration Across Nerve Gaps

When the gap between the nerve stumps is too long, anastomosis is not possible because the nerve cannot be stretched to extend across the gap. Therefore, to restore function to such nerves, the gaps must be bridged with a material that both supports and promotes axon regeneration entirely across the gap.

### 2.7. Autografts

It was originally hypothesized that the best material for inducing axons to regenerate across nerve gaps would be a length of the autologous peripheral nerve [41]. The most commonly used donor nerves are the cutaneous saphenous, medial antebrachial cutaneous, and sural nerves [42,43,44,45].

The following sections discuss the efficacy of sensory nerve grafts and other techniques in promoting axon regeneration and neurological recovery.

## 3. Sensory Nerve Grafts: Limitations

### 3.1. Loss of Sensory Nerve Function

The primary drawback to using lengths of nerve as a graft is that their use requires sacrificing the function of that nerve. This creates a permanent neurological deficit [15,46,47].

### 3.2. Incorrect Schwann Cell Phenotype

Both sensory and motor nerve grafts have been tested for their efficacy in promoting axon regeneration and restoration of function. Motor nerve grafts induce significantly greater axon regeneration than sensory nerve grafts. This is because sensory and motor nerve Schwann cells express distinctly different phenotypes, and each best supports the regeneration of their specific axon phenotype [16,48]. Although motor nerve grafts are more effective than sensory nerve grafts in promoting axon regeneration across a nerve gap, they are not used because it is considered unethical to sacrifice a motor nerve function, when the loss of a pure sensory nerve has minimal impact on the patient.

### 3.3. Inflammation

The standard technique for securing nerve grafts in place is to use multiple sutures through the epineurium of the graft and the nerve stumps. However, sutures often cause inflammation and scarring, both of which inhibit axon regeneration [49]. This problem can be overcome by placing a degradable collagen [50] or fibrin [51] conduit around the site of nerve stump anastomosis, which stabilizes the juxtaposition of the nerve stumps. An alternative technique is to apply fibrin glue to the anastomosis site of the nerve stumps [52].

### 3.4. Necrosis

Sensory nerves generally have a smaller diameter than the mixed sensory/motor nerves they are commonly used to repair. Often, multiple small diameter grafts are used so that the final diameter of the grafts approximates that of the nerve to be repaired. However, smaller diameter grafts are correlated with less functional recovery than larger diameter grafts [53,54,55]. This is ascribed to the lack of vascularization leading to necrosis of the Schwann cells within the graft [54,56]. Necrosis reduces, if not blocks, axon regeneration through the graft. This situation is not improved when multiple small grafts are used. To avoid necrosis, the best approach is to use vascularized nerve grafts (see the following section on vascularized grafts and inducing vascularization).

### 3.5. Decreasing Recovery with Increasing Gap Length

Sensory nerve grafts promote good to excellent functional recovery only when nerve gaps are <2 cm in length [57,58,59]. The extent of recovery decreases to only good for gaps ≥ 3 cm in length [60,61], decreases further for gaps up to 4 cm [58,62,63], and there is a precipitous decrease in recovery for gaps > 4–5 cm [58,64,65]. Few axons regenerate across grafts of 8 cm in length [57,60] and there are no reports of axons regenerating across gaps >10 cm in length [7,58,63,65,66,67]. Thus, neurological recovery decreases with increasing gap length [53]. Therefore, sensory nerve grafts are only considered reliable for “short” nerve gaps (≤3 cm) [55,68]. Furthermore, no material is Food and Drug Administration (FDA)-approved for repairing nerve gaps >3 cm in length [55,69,70].

The reduction in axon regeneration across long nerve grafts appears correlated with the longer time required to vascularize longer grafts [71]. As mentioned above, without vascularization, the graft environment is ischemic, which inhibits axon regeneration [72].

### 3.6. Decreasing Recovery with Increasing Time between Nerve Injury and Repair

Anastomosing nerve stumps of non-traumatic transected radial nerves of young males (25 years) immediately [73] or within 14 days [7] of the injury generally results in good neurological recovery for 67% of subjects [73]. However, as the delay in performing the repair increases, the recovery of good function decreases to 30%, fair in 28%, and fails for 42% of patients [7,73].

Similarly, when short nerve gaps are repaired using sensory nerve grafts, recovery is very good to excellent following repairs performed ≤ 14 days post-trauma [7] or good to excellent for repairs performed ≤ 2 months post-trauma [74]. However, the extent of recovery decreases significantly for repairs performed > 3 months post-trauma [62,75,76]. Thus, delays of >2 months result in good recovery in only 49% of patients [62], but are poor for repairs performed > 6 months post-trauma [62,77,78,79]. No recovery is reported for repairs performed > 10 months post-trauma [80,81].

### 3.7. Schwann Cell Senescence

Axon regeneration across long nerve gaps requires Schwann cells of the proximal nerve stump to proliferate extensively so they can both promote and accompany the axons as they regenerate. The limited regeneration of axons across long nerve gaps, and when nerves are repaired at long times post-trauma, is in part, attributed to Schwann cell senescence [69]. Senescence is associated with increased expression of markers for β-galactosidase, p16^INK4A^, and interleukin (IL)6 [82]. Thus, over time, without contact with an axon, Schwann cells lose their ability to proliferate, synthesize, and release neurotrophic factors, which are required to promote axon regeneration [82,83].

### 3.8. Neuron Loss of Ability to Regenerate

Another explanation for the significant decrease in axon regeneration with increasing time of motor neuron axotomy is that many motor neurons lose the ability to extend axons [83]. The decrease in the ability of motor neurons to regenerate with increasing time of axotomy appears to be associated with the downregulation of neuregulin 1, which is required for axon regeneration [38,84,85]. However, it is important to note that while some motor neurons lose this capacity, others can regenerate even after many years of axotomy. In a clinical case, motor neurons that were axotomized for 22 years were able to extend axons to reinnervate newly denervated muscles [86].

It is not known what determines which motor neurons retain or lose the capacity to regenerate. However, understanding this regulation might provide insights into how neurons’ gene expression can be modulated to promote enhanced axon outgrowth. Thus, it would be interesting to compare the gene expression between motor neurons with and without the capacity to regenerate after prolonged axotomy to determine the gene expression that underlies the capacity to regenerate normally. As is discussed later, techniques have been developed that induce motor neurons, which appear to have lost the ability to regenerate, to extend axons.

### 3.9. Decreasing Recovery with Patient Age

The recovery of function following nerve graft repair is best for patients < 20–25 years of age [87], and the extent of recovery decreases significantly with increasing patient age [87,88]. This is, in part, attributed to the decreasing capacity of sensory nerve grafts to promote axon regeneration (i.e., their Schwann cells becoming senescent) [69]. This change is also associated with the downregulation over time post-axotomy of the ability of neurons to synthesize and release neuregulin 1, which is required for axon regeneration [38,84,85].

Endoneurial vasculature is required for the outgrowth of axons from the proximal stump, and angiogenesis is essential for nerve regeneration [89]. However, clinically, with increasing age, there is a reduction in, or lack of, angiogenesis in response to injury [90]. This change is due to the decrease in nerve injury-induced upregulation of the expression and release of vascular endothelial growth factor (VEGF) [91,92]. VEGF is required for inducing vascularization, which, in turn, is required for axon regeneration. This suggests that vascular abnormalities might play a role in the decreasing ability of axons to regenerate with increasing age. This further suggests that promoting vascularization may increase the extent of neurological recovery in older patients.

### 3.10. Enhancing the Efficacy of Autografts to Promote Axon Regeneration

The extent and distance axons regenerate decreases with increasing graft length, increasing delays between nerve injury and repair, and increasing age. As already stated, these limitations are associated with (1) the inability of neurons to extend axons, (2) development of Schwann cell senescence, and (3) the failure, or slow process of, graft re-vascularization. Despite these challenges, various techniques overcome these limitations and promote axon regeneration.

Axon regeneration can be triggered from neurons that no longer extend axons, and the rate and extent of axon regeneration increases by refreshing central nerve stumps and then stimulating them electrically, or by applying neurotrophic factors [93,94,95]. These techniques restore the capacity of neurons to regenerate [96,97,98] while also inducing the senescent Schwann cells of the distal portion of the nerve to proliferate and release neurotrophic factors that promote the extension of axons from long-term axotomized neurons [99,100].

## 4. Electrical Stimulation

### 4.1. Promoting Axotomized Neurons to Extend Axons

As mentioned previously, within increasing the time of axotomy, neurons lose their ability to extend axons [83]. However, these neurons can be induced to extend axons by electrical stimulation of the proximal portion of the transected nerves for as little as one hour [94,101,102] This results in a 34–50% increase in the number of neurons that extend axons [98,103,104]. Electrical stimulation also induces a 2.3-fold increase in the extent of axon sprouting from transected axons [105] while increasing the speed of axon regeneration [101,103,106]. At the same time, in animal models, electrical stimulation increases the distance axons regenerate across nerve gaps, the accuracy of sensory vs. motor axon innervation of their appropriate targets, and extent of functional recovery [107,108,109,110]. Electrical stimulation of peripheral nerve clinically also induces enhanced axon regeneration [102].

Electrical stimulation acts by inducing neurons to upregulate their level of cyclic-AMP [89,103]. This, in turn, induces motor neurons to upregulate their expression and synthesis of the neurotrophic factor BDNF and its trkB receptor mRNA, as well as the mRNA for other factors that enhance axon regeneration [99,111] These actions make neurons more receptive to regeneration-promoting factors [112].

### 4.2. Activating Senescent Schwann Cells

The decrease in axon regeneration across long nerve grafts, and with increasing time between nerve injury and repair, is attributed in part to Schwann cells becoming senescent when they lose contact with axons [69,75]. Thus, they stop proliferating and releasing neurotrophic factors that are required to promote axon regeneration. Electrical stimulation enhances axon regeneration by inducing the senescent Schwann cells to proliferate, migrate, and upregulate their synthesis and release of neurotrophic factors, which act to promote axon regeneration [99,113].

Schwann cells can also be induced to exit their senescent state by the application of VEGF [76,112,114] and marrow-derived mesenchymal stem cells. Thus, the Schwann cells reinitiate their ability to proliferate and to express and release neurotrophic factors [80].

### 4.3. Vascularized Nerve Grafts and Promoting Vascularization

The standard sensory nerve graft is cut from a donor nerve without maintaining its vasculature. As stated earlier, non-vascularized grafts become necrotic, which creates a toxic environment that inhibits axons regeneration until re-vascularization occurs. However, re-vascularized takes days to develop, and takes more time for longer nerve grafts. This is because vascularization normally progresses from one end of the graft to the other.

The limitation of using non-vascularized grafts is avoided by using vascularized nerve grafts. In the rat sciatic nerve model, vascularized grafts induce significantly greater neurological recovery than non-vascularized nerve grafts [115]. Clinically, vascularized nerve grafts are required for axons to regenerate across gaps of longer than 6 cm [116]. Although vascularized nerve grafts are more effective in restoring function than non-vascularized grafts, they are not commonly used because the surgery is more complicated and time-consuming.

An alternative technique to using vascularized autografts is to induce the rapid re-vascularization of non-vascularized grafts. This can be done by pre-treating nerve grafts with VEGF before using them [117]. This treatment stimulates neovascularization of the graft, and Schwann cell invasion into the graft [117] and can reduce the time of graft ischemia by three days [118].

Axon regeneration can also be enhanced by using autografts with cells overexpressing VEGF, which leads to hyper-vascularization [119]. This enhances axon regeneration by reducing endoneurial scarring, by maintaining the viability of Schwann and other cells within the graft, and by decreasing fibroblast infiltration. This results in a good nutritional environment for supporting axonal regeneration.

The decrease in axon regeneration through nerve grafts with increasing age is also attributed to reduced graft vascularization. This is because aging is associated with a decrease in the upregulation of the expression, and release of VEGF following nerve injury [91]. Following nerve injury to aged mice, there is a significant reduction in the upregulation of VEGF synthesis and release, and thus, a failure of axons to regenerate. These findings suggest that, with increasing age, vascular abnormalities might play a role in the decreasing ability of axons to regenerate. They also suggest that clinically, inducing enhanced vascularization might enhance axon regeneration and functional recovery.

### 4.4. Conclusion about Sensory Nerve Grafts

The use of sensory nerve grafts leads to a permanent neurological deficit of the donor nerve. While such grafts can induce good to excellent functional recovery, such recovery is only for young patients [120,121], short gaps [65,67] (Aszmann et al., 2008b), and 3 when the repairs are performed within a short time post-trauma [77,79,81]. As any one of these values increases, the extent of recovery decreases precipitously. When the values of two or all three of these parameters increase, there is minimal to no functional recovery. However, even when nerve repair surgery is offered to patients who are considered good candidates for recovery function, <50% of them recover function [122]. Thus, most individuals who suffer peripheral nerve traumas suffer permanent neurological deficits and commonly chronic neuropathic pain. Therefore, there is a need for novel techniques that induce recovery under conditions where sensory nerve grafts are not effective.

The following sections examine other techniques that have been tested for their efficacy in enhancing the extent of axon regeneration and neurological recovery across nerve gaps. They also examine methods tested for increasing their efficacy in enhancing the extent of axon regeneration. The final section briefly discusses a novel technique that holds promise for restoring function, even when simultaneously the values of all three nerve injury parameters far exceed those when sensory nerve grafts are effective in promoting axon regeneration.

## 5. Allografts

Conceptually, a good alternative to autographs is cellular cadaveric allografts. First, they would avoid the need to sacrifice a sensory nerve function; second, they provide both a three-dimensional extracellular matrix for supporting and promoting axon regeneration and Schwann cells, which can release neurotrophic factors for promoting axon regeneration. However, using allografts requires the administration of immunosuppressive drugs to avoid graft rejection and regeneration failure [123], but immunosuppressants are associated with significant clinical morbidity [124]. They also cause unwanted side effects, such as suppressing the regeneration-promoting capacity of host Schwann cells [30]. Thus, the use of cellular allografts for peripheral nerve repair is rare and their uses are limited to the most severe cases of nerve injuries, such as those involving repairing long nerve gaps [125].

An alternative to cellular allografts is to use acellular allografts. These can be used without immunosuppression after eliminating their immunogenicity [126]. Although they lack cells, they typically maintain a highly organized extracellular matrix scaffold, which can induce axon regeneration. Acellular nerve allografts, also called processed nerve allografts, are now increasingly used instead of autografts [124,127,128,129,130,131].

In a comparative study bridging 1.4-cm sciatic nerve gaps in rats, acellular allografts, isografts, and empty collagen conduits were seen to induce similar axon regeneration entirely across the gap [126]. Another recent comparative clinical study looked at the success of sensory recovery when digital nerve gaps of 1.4 and 1.8 cm were bridged by acellular nerve grafts and empty collagen tubes. Allografts vs. collagen tubes induced excellent outcomes in 39% vs. 48%, good in 55% vs. 26%, and poor in 6% vs. 26%, respectively [126]. However, for 2.8-cm gaps, isografts were more effective than allografts, but conduits were not effective [120]. Other studies determined that acellular allografts are excellent for promoting axon regeneration across gaps 1–1.5 cm [31], 2–3 cm [131], and up to 5 cm meaningful recovery across gaps [132]. However, similar to autografts, their efficacy decreases with increasing gap length [53] and they are not recommended for (or FDA-approved) use across “long” nerve gaps, considered to be >3 cm in length [55,68,82,125,133,134,135].

### Enhancing the Regeneration-Promoting Capacity of Allografts

The efficacy of acellular allografts in promoting axon regeneration and functional recovery can be increased by infusing them with neurotrophic factors, such as glial-derived neurotrophic factor (GDNF) [136], nerve growth factor (NGF) [24], BDNF plus ciliary neurotrophic factor (CNTF) [137], VEGF [138], and βNGF and VEGF [139].

Additional techniques for enhancing axons regeneration through long allografts grafts include filling them with platelet-rich plasma (PRP) [140], autologous Schwann cells [141,142], adipose-derived mesenchymal stem cells (ADSCs), or primary Schwann cell-like differentiated bone marrow-derived mesenchymal stem cells (DMSCs) [142,143,144,145]. These cells induce enhanced axon regeneration by releasing neurotrophic factors [146,147].

Although acellular allografts and autografts induce similar extents of axon regeneration across short nerve gaps, acellular allografts are less effective when used for long nerve gaps [148]. However, when the efficacy of acellular allografts is enhanced, they induce axon regeneration that is comparable to that induced by autografts. Unfortunately, the techniques required to enhance the efficacy of acellular allografts have only been tested in animal models, and none can presently be applied clinically. Therefore, sensory nerve grafts remain the “gold standard” for bridging long nerve gaps.

## 6. Nerve Conduits

### 6.1. Conduit Composition

Regardless of their composition, empty conduits longer than 1 cm in length, generally induce poor, if any, axon regeneration. However, conduit composition is important. Silicon conduits induce axon regeneration [149,150] but can have the complication of rigidity, which can result in significant chronic nerve compression and irritation at the implantation site [151]. In this case, they may have to be removed, which endangers any recovered neurological function. Therefore, it is advisable to use a fully degradable matrix that does not negatively affect axon regeneration [152].

Axon regeneration is extensive through conduits composed of fibrin [145,153], hydrogel tubes [154], alginate/chitosan polyelectrolyte [155,156,157], poly epsilon-caprolactone [158,159], polyglycolic acid [160], poly(lactic-co-glycolic acid) or silk-based [161,162,163,164]. Conduits composed of decellularized human umbilical artery [165] and muscles [166] are also effective in promoting axon regeneration across nerve gaps.

Veins are promising as conduits [167,168]. Clinically, empty vein conduits appear to have a 3-cm-long limit for their ability to induce axon regeneration [169]. A meta-analysis of published papers determined that for nerve gaps up to 4 cm in length, vein conduits did not induce any significant improvement in sensory recovery outcome compared to conduits of other materials [170]. However, the axon regeneration-promoting efficacy of vein conduits is enhanced when they are filled with PRP [171,172,173], muscle [174], pre-degenerated muscle [175,176], muscle seeded with neural-transdifferentiated human mesenchymal stem cells [177], and minced peripheral nerves [178]. One study in sheep showed that using the median epineural sheath as a conduit can restore median nerve function across 6-cm-long nerve gaps [179].

### 6.2. Conduit Architecture

Electrospun collagen/poly(lactic-*co*-glycolic acid) (PLGA) conduits with a three-dimensional internal structure induce more extensive axon regeneration than conduits without a three-dimensional structure [180]. Functional recovery through conduits filled with pure fibrin gel is significantly increased when three-dimensional collagen tubes are filled with gelatin containing biodegradable poly-epsilon-caprolactone and collagen/ poly ε-caprolactone (PCL) sub-micron scale fibers [181].

### 6.3. Conduits Containing Neurotrophic and Other Factors

The efficacy of collagen conduits is increased when they contain, or release, neurotrophic or other factors, such as GDNF [180] or neurotrophin-3 (NT-3) [182]. Alginate/chitosan conduits induce more extensive axon regeneration when they contain or release simvastatin [183], NGF [184,185,186,187], GDNF [184], VEGF [188], GDNF and NGF [24,189], or pleiotrophin [15]. Efficacy is also increased when alginate/chitosan conduits are combined with fibronectin, laminin [190], or when hydrogel conduits contain Matrigel, collagen, heparin (sulfate), laminin, or fibronectin [190]. Conduit efficacy is further increased by the release of neurotrophic factors within conduits from biodegradable polymeric with aligned heparin-conjugated nanofibers [191,192].

Although pure fibrin-filled conduits induce axon regeneration [153], this influence is increased by adding neurotrophic and other axon regeneration-promoting factors [59]. This increase in efficacy is due to the fibrin binding the growth factors, such as basic fibroblast growth factor (bFGF) [65], NGF, BDNF, and NT-30 and factors from the PDGF/VEGF, FGF, and tumor growth factor-beta (TGF-β) families [73]. Fibrin also facilitates the promotion of axon regeneration by binding Schwann cell-released extracellular matrix factor laminin [59]. The binding of these factors converts the fibrin from a passive to a potent regeneration-promoting three-dimensional matrix. This influence is supported by data showing that the application of neurotrophic factors within fibrin glue to the sites of nerve stump anastomosis significantly increases axon regeneration compared to applying the factors directly without fibrin [73,193]. Regeneration is also enhanced by infusing conduits with FK506, an immunosuppressive agent [109,194].

### 6.4. Conduits Containing Cells

The efficacy of fibrin conduits is enhanced when they are filled with autologous undifferentiated adipose-derived stem cells [145] or contain mesenchymal stem cells [195], and when hydrogel conduits contain mesenchymal stem cells [196]. The efficacy of chitosan conduits is enhanced when they contain combinations of fibronectin and laminin with mesenchymal stem cells (MSCs) or Schwann cells [190], chitosan/PLGA scaffolds are combined with mesenchymal stem cells [197], and when three-dimensional alginate/chitosan conduits are filled with muscle fibers [198]. Adding PRP to the inside of silicon tubes bridging nerve gaps increases the extent of axon regeneration compared to that induced by empty silicon conduits [149,198,199]. Axon regeneration through conduits can be enhanced by adding olfactory ensheathing cells [200] or dissociated Schwann cells, which release axon regeneration-promoting neurotrophic factors, such as NGF, BDNF, NT-3, CNTF, GDNF, and cell adhesion molecules (CAMs) [175] and by building up a basement membrane [69].

Clinically, the number of axons and the distance they regenerate, are significantly increased by adding minced pieces of peripheral nerve to empty conduits [184]. Clinically, bridging a 5-cm-long radial nerve gap with two sensory nerve grafts within a pure fibrin-filled collagen tube induces excellent sensory and motor recovery across a 5-cm-long nerve gap [201].

### 6.5. Conduit Composition and Electrical Stimulation

Neurite outgrowth from neuron-like cells in vitro is enhanced when they are grown on nerve guidance channels composed of an electrically conductive polymer (oxidized polypyrrole) [202]. Electrically conductive biodegradable polymer composite materials enhance the rate of neurite outgrowth from cultured PC12 cells [101]. Similarly, electrical stimulation of stem cells seeded onto electrospun conducting polymer nanofibers induces neurite extension [203]. Thus, electrical stimulation of conduits composed of such material may enhance axon regeneration in vivo.

### 6.6. Conclusion for Conduits

Axon regeneration through conduits can be increased compared to that through empty collagen conduits by modifying their composition, architectures, and the factors/cells they contain. However, no conduit tested is as effective as allografts in inducing axon regeneration across both short and long gaps [55,68,204] under nerve injury conditions that are common clinically, such as a long gap in older patients that must be repaired at a long time post-trauma. An additional challenge is that the conduits that are most effective in animal models cannot be used clinically. Therefore, collagen conduits are the most commonly used clinically.

## 7. Novel Technique

A novel technique was tested clinically for bridging a nerve gap. It involved bridging a 12-cm-long ulnar nerve gap with a PRP-filled collagen tube [205]. The subject was a 58-year-old male, and the repair was performed post-nerve trauma (Figure 1). The subject recovered good motor and sensory function [205]. Thus, both sensory and motor function can be restored, even when simultaneously, all three nerve trauma values far exceeded those where sensory nerve grafts alone are effective. This result shows that functional recovery can be restored under conditions where sensory nerve grafts, allografts, and conduits are not effective. The technique is presently being further tested.

Support for the clinically observed efficacy of a PRP-filled collagen tube enhancing axon regeneration comes from both a clinical study and a number of animal studies. Clinically, the application of PRP eye drops to the cornea enhances the regeneration of sensory innervation [206]. PRP applied to the site of anastomosed rat nerve stumps enhances the extent of axon regeneration in rats [207,208,209] and guinea pigs, In rabbits, PRP applied to an autograft increases Schwann cell proliferation and the extent of axon regeneration [210]. Bridging a rat sciatic nerve gap with a PRP-filled silicon tube [198,211] or surrounding the sciatic nerve anastomosis site with a PRP-saturated membrane [212] enhances axon regeneration. In rabbits, filling vein grafts with PRP induces significant axon regeneration compared to empty vein grafts [198,206,213,214,215].

The influence of PRP in enhancing axon regeneration has been proposed to result from platelet-released neurotrophic and other factors [214,215,216,217]. An additional platelet potential contributor to enhanced axon regeneration is VEGF, which, as discussed earlier, induces enhanced axon regeneration by inducing rapid vascularization of the entire nerve gap [218,219]. This result suggests that platelet-released neurotrophic and other factors create an environment within the collagen tube that promotes axon regeneration despite a long nerve gap, a long delay between nerve trauma and repair, and an old patient. These results suggest that the platelet-released factors play a number of roles to induce axon regeneration and functional recovery: (1) They induce neurons to extend axons long after they normally do not; (2) induce the Schwann cells of the proximal nerve stump to proliferate, and release axon regeneration-promoting factors; (3) induce the Schwann cells of the proximal stump to migrate with the elongating axons; (4) induce the Schwann cells of the distal nerve to proliferate and release neurotrophic factors and support axon regeneration into and through the distal nerve to the denervated targets; (5) a platelet-released factor, potentially VEGF, promotes vascularization of the gap region.

Confirmation of this result will mean that, without sacrificing a sensory nerve function, neurological recovery can be restored under conditions of long nerve gaps, when repairs are performed years post-trauma and to older patients.

## 8. Conclusions

Sensory nerve grafts, allografts, and conduits all induce axons to regenerate across nerve gaps. Acellular allografts and conduits are considered to induce neurological outcomes only for nerve gaps <6 cm, although they are routinely used with reliability for “short” nerve gaps considered to be <3 cm, and allografts are FDA-approved only for repairing nerve gaps > 3 cm in length. All three techniques suffer the same imitations of decreasing efficacy with increasing gap length, increasing time between nerve trauma and repair, and increasing patient age. Although the distance across which allografts and conduits induce axon regeneration can be increased by modifying them in various ways, none of those techniques can be used clinically. Therefore, despite their limitations, sensory nerve grafts remain the clinical “gold standard” for repairing peripheral nerves [69,70,166,220,221,222,223]. The recovery of function across a 12-cm-long nerve gap of a 58-year-old patient, repaired 3.25 years post-trauma, suggests that functional recovery can be established, even when the values of all three injury parameters far exceed those where autografts are effective. Further testing and development of the technique are required to determine its reliability and the limits of its efficacy.

## Figures and Tables

**Figure 1 ijms-21-01808-f001:**
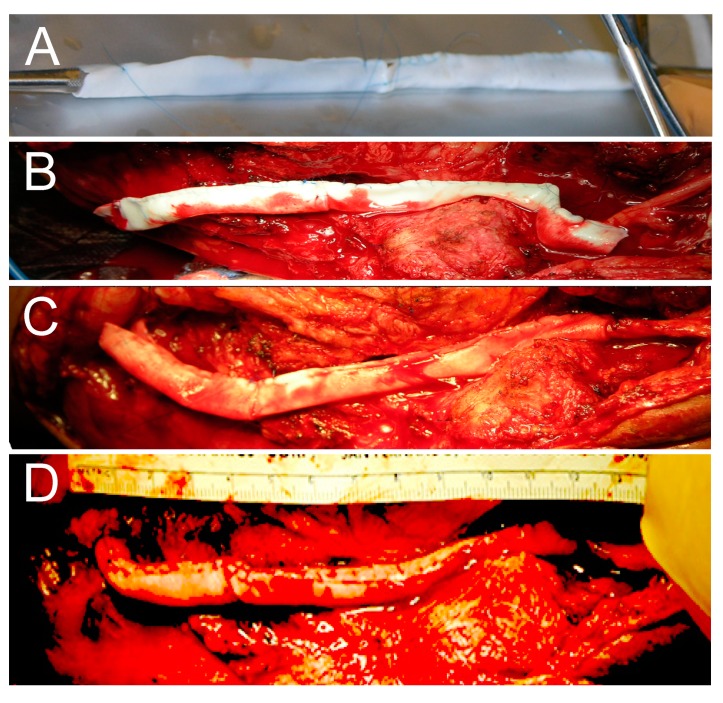
Repairing an ulnar nerve with a 12-cm-long gap. (**A**) Sewing two 4 × 8 cm collagen sheets together end to end, and then into a 16-cm-long tube around the handle of a surgical tool. (**B**) The collagen tube cut to a 12.6 cm length and placed in the nerve gap. (**C**) The proximal and distal nerve stumps secured about 3 mm into the collagen tube. (**D**) Completed nerve gap repair with the collagen tube filled with autologous platelet-rich plasma (PRP).

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
