# Peer review of "Restoration of Neurological Function Following Peripheral Nerve Trauma"

_ijms, 2020, doi:10.3390/ijms21051808_

Round 1

Reviewer 1 Report

In this review the authors examines the efficacies and limitations of sensory nerve grafts and various other techniques used to induce functional neurological recovery, and how these might be improved to induce more extensive functional recovery. The study is well documented and scientifically satisfactory. The bibliography is very good.

Author Response

The reviewer complemented the manuscript. but made no recommended changes.

Reviewer 2 Report

This manuscript covers an important topic – “examine the limitations of sensory nerve grafts, and other techniques that have been tested for their efficacy in promoting axon regeneration and neurological recovery across nerve gaps. It also examines how it might be possible to increase the efficacy of some of these techniques. Finally, it presents preliminary data from a clinical study that suggest neurological recovery might be possible, even when all three nerve injury parameters far exceed those when sensory nerve grafts are effective”. The strengths of the paper include the public health importance of the topic and the comprehensive data source. However, the paper could be substantially strengthened by addressing the following concerns:

-In general, English language edition needs to be considered.

- It would be easier to understand if before starting to discuss about limitation, briefly mentioning about the technique and then continue with the appropriate subheadings.

-One of the limitations of SENSORY NERVE GRAFTS is necrosis, however the discussion under this subheading is about comparison between multiple nerve grafts and single graft.

-Some of the factors were mentioned are the limitation and some of them are the side effects and the factors that influence the outcome of the applied technique.

-Regarding “Influence of increasing time between nerve injury and repair on the extent of recovery”, summarizing on the time point between trauma and repair would enhance comprehension of the discussion. Moreover, how important age affect the recovery in connection with time point? Does gender have an effect as well?

- What is the function of VEGF in axon regeneration following acellular allografts? It needs more detail, since it is vascular growth factor and has different subtypes.

-Authors mentioned “allografts are never as effective as autografts in promoting neurological recovery.” While they mentioned “Acellular allografts are considered excellent for promoting axon regeneration across gaps 1 - 1.5 cm,[15] 2 - 3 cm,[18] and even 7 cm.[68]”

-Line 218 needs to be revised.

- NERVE CONDUITS section needs to have summarize at the end of section. It included information of using several materials with advantages and disadvantages.

- It is better “Enhancing the efficacy of autografts” be as a part of SENSORY NERVE GRAFTS, since it is related to this section. Otherwise it is confusing and makes it difficult to follow the section.

- Under” Enhancing the efficacy of autografts” section, more detail about the mechanistic role of VEGF is needed.

-Adding a table as a summary of previous studies will give comprehensive information regarding different methods of nerve repair following trauma.

-Authors did not mention how type of injury may influence the axonal regeneration by application of these therapeutic methods? Electrical Stimulation

-How about Electrical Stimulation method comparing with the therapeutic method which were mentioned in this review? It would be good to add some information in this regard.

-It would be great to add section regarding classification of nerve injuries and discuss about it in connection restoration of neurological function.

-Authors added image of the novel method they applied for the axon regeneration and functional recovery. And they mentioned as a preliminary result and suggested as the possible developed method of axonal regeneration. For such this conclusion and taking it as a result, it is needed to include several factors such as number of patients, demographic, follow up, neurological examination, proper statistical test. Accordingly, it can not be used in this shape as a preliminary result. This part should be removed from the manuscript or add all needed information.

Author Response

Detailed discussion of recommended changes, additions, and author responses

  1. Before starting to discuss limitation, briefly discuss the technique and then continue with the appropriate subheadings.

This has been done by starting the paper with a discussion of the need for the use of nerve grafts and their efficacies. This is followed by a discussion of the limitations of these nerve grafts.

  1. One of the limitations of SENSORY NERVE GRAFTS is necrosis, however the discussion under this subheading is about comparison between multiple nerve grafts and single graft.

This section has been changed to more clearly focus on the problem of necrosis within nerve grafts and how it blocks axon regeneration. It also discusses how multiple small grafts increase the complications of necrosis, and their use is ill-advised.

This modification resulted in the introduction of the need for the use of vascularized grafts, or the rapid re-vascularization of non-vascularized nerve grafts to reducing necrosis.

  1. Some of the factors mentioned are limitation and some are the side effects and the factors that influence the outcome of the applied technique.

This complication has been addressed in clearer discussions of each point.

  1. Regarding “Influence of increasing time between nerve injury and repair on the extent of recovery”, summarizing on the time point between trauma and repair would enhance comprehension of the discussion.

The authors believe the influences of increasing time between nerve trauma and repair on the extent of neurological recovery was extremely clearly stated in the first paragraph under the heading of deceasing recovery with increasing delays between trauma and repair in original submission.

The authors increased the discussion of the potential mechanisms that underlie this effect.

The authors do not believe any additional material, discussion, or table are needed.

  1. How important is increasing age on the extent of recovery of function?

The authors believe this issue is extremely clearly expressed in the first paragraph under the heading changes in recovery with increasing age. As was written in the original submission

The authors increased the discussion of the potential mechanisms that underlie this effect.

The authors do not believe any additional material, discussion, or table are needed.

  1. Does gender affect the extent of axon regeneration?

A new section entitled Gender and Age has been added, in which the influences of gender and patient age on axon regeneration are discussed. Those sections also discuss the potential bases for these influences.

  1. What is the function of VEGF in axon regeneration following acellular allografts? It needs more detail

This was an extremely important recommendation. A new section had been added entitled Vascularization. This does into details about the importance of VEGF and vascularization of nerve grafts, the use of vascularized nerve grafts, and techniques that induce enhanced vascularization of nerve grafts.

  1. -Authors mention that “allografts are never as effective as autografts in promoting neurological recovery.” While they mentioned “Acellular allografts are considered excellent for promoting axon regeneration across gaps 1 - 1.5 cm,[15] 2 - 3 cm,[18] and even 7 cm.[68]” Therefore, Line 218 needs to be revised.

The authors must clarify that allografts have been reported to be almost as effective as autografts. However, that is not generally the case. Rather, allografts and autografts are both considered to induce similar axon regeneration across nerve gaps up to about 3 cm, but for longer gaps, autografts are more effective.

In the section on improving the efficacy of allografts, such as by adding cells or factors, it is stated that these modifications allow allografts to act comparably to autografts. However, it is also mentioned that these efficacies have only been shown in animal models, and none of those techniques can presently be use clinically.

  1. NERVE CONDUITS section needs to have summarize at the end of section and to discuss the advantages and disadvantages of using several materials.

The section on nerve conduits mentions many of the different materials that have been used to construct conduits. It discusses how the composition of the conduit, its 3-dimentional structure, the addition of various types of different cells and different factors, and electrical affect axon regeneration.

The authors do not believe a more detailed discussion of the efficacies of each specific material used to construct nerve conduits is required.          

  1. “Enhancing the efficacy of autografts” should be moved to SENSORY NERVE GRAFTS, since it is related to this section.

This material has been moved.

  1. Under” Enhancing the efficacy of autografts” section, more detail about the mechanistic role of VEGF is needed.

A new section discussing VEGF has been added.

In addition, as stated earlier, an attempt has been made to stress, wherever it is required, the important role of VEGF and vascularization in promoting axon regeneration. This is now more clearly raised and discussed in the sections on long nerve grafts, delay in repairs, and that along with patient aging there is a reduction in the injury-induced vascularization responses.

  1. Add a table to summarize the all the previous studies on nerve repairs.

The authors do not believe a summary of previous studies is necessary. This is because all techniques suffer the same limitations: reduced neurological recovery with increasing gap length, delay between trauma and repair, and patient age.

Further, at the end of each section discussing nerve grafts, allografts, and conduits, there is a summary of the efficacies of those approaches. An additional summary is not needed.

  1. Authors do not mention how type of injury may influence the axonal regeneration

A new section has been added entitled Type of nerve trauma. This discusses how different types of nerve injury influences neurological recovery.  

  1. Discuss how Electrical Stimulation influences recovery

The original paper discussed how electrical stimulation of nerves can be used to enhance the extent of axon regeneration and recovery. A new section has been added entitled ELECTRICAL STIMULATION. This discusses in greater detail the results of electrical stimulation on axon regeneration. It also mentions how electrical stimulation is effective not only in animal models, but clinically.

Note: There is also mention of how electrical stimulation via electrically conduit conduits may enhance axon regeneration.

  1. It would be great to add section regarding classification of nerve injuries and discuss about it in connection restoration of neurological function.

This review focuses on techniques being used and developed in attempts to increase the extent of functional recovery to nerves with a gap. The authors do not believe a discussion of how different techniques influence axon regeneration following different types of nerve injuries is necessary, or would be beneficial. In addition, there is no body of work to discuss in which the efficacies of different types of techniques have been compared for restoring function to nerve following different types of nerve injuries.

  1. Authors discuss a preliminary result and suggested as the possible developed method of axonal regeneration. However, they do not provide sufficient information to support considering the data as a result, and therefore, cannot call it a preliminary result. Therefore, this should be removed from the manuscript or add all needed information.

The reviewer is entirely correct. Therefore, a description of that new technique has been eliminated. Instead, the discussion had been changed to discuss the results of the clinical application of another similar technique, and a figure has been added. This technique restored function under conditions were no other technique, including sensory nerve grafts is effective. This is material is from a published case report (by one of the authors) that has been peer reviewed. The figure is new and no approval to use it is required.

Round 2

Reviewer 2 Report

The manuscript is revised in a satisfactory manner. 

This manuscript is a resubmission of an earlier submission. The following is a list of the peer review reports and author responses from that submission.